# Acute Modification of Hemodynamic Forces in Patients with Severe Aortic Stenosis after Transcatheter Aortic Valve Implantation

**DOI:** 10.3390/jcm12031218

**Published:** 2023-02-03

**Authors:** Alessandro Vairo, Lorenzo Zaccaro, Andrea Ballatore, Lorenzo Airale, Fabrizio D’Ascenzo, Gianluca Alunni, Federico Conrotto, Luca Scudeler, Daniela Mascaretti, Davide Miccoli, Michele La Torre, Mauro Rinaldi, Gianni Pedrizzetti, Stefano Salizzoni, Gaetano Maria De Ferrari

**Affiliations:** 1Division of Cardiology, Cardiovascular and Thoracic Department, Citta della Salute e della Scienza Hospital, 10126 Turin, Italy; 2Internal Medicine and Hypertension Division, Department of Medical Sciences, Città della Salute e della Scienza di Torino, University of Turin, 10126 Turin, Italy; 3Division of Cardiac Surgery, Department of Surgical Sciences, Città della Salute e della Scienza di Torino, University of Turin, 10126 Torino, Italy; 4Department of Engineering and Architecture, University of Trieste, 34127 Trieste, Italy

**Keywords:** TAVI, echocardiography, hemodynamic forces, predictors

## Abstract

Transcatheter aortic valve implantation (TAVI) is the established first-line treatment for patient with severe aortic stenosis not suitable for surgery. Echocardiographic evaluation of hemodynamic forces (HDFs) is a growing field, holding the potential to early predict improvement in LV function. A prospective observational study was conducted. Transthoracic echocardiography was performed before and after TAVI. HDFs were analyzed along with traditional left ventricular (LV) function parameters. Twenty-five consecutive patients undergoing TAVI were enrolled: mean age 83 ± 5 years, 74.5% male, mean LV Ejection Fraction (LVEF) at baseline 57 ± 8%. Post-TAVI echocardiographic evaluation was performed 2.4 ± 1.06 days after the procedure. HDF amplitude parameters improved significantly after the procedure: LV Longitudinal Forces (LF) apex-base [mean difference (MD) 1.79%; 95% CI 1.07–2.5; *p*-value < 0.001]; LV systolic LF apex-base (MD 2.6%; 95% CI 1.57–3.7; *p*-value < 0.001); LV impulse (LVim) apex-base (MD 2.9%; 95% CI 1.48–4.3; *p*-value < 0.001). Similarly, HDFs orientation parameters improved: LVLF angle (MD 1.5°; 95% CI 0.07–2.9; *p*-value = 0.041); LVim angle (MD 2.16°; 95% CI 0.76–3.56; *p*-value = 0.004). Conversely, global longitudinal strain and LVEF did not show any significant difference before and after the procedure. Echocardiographic analysis of HDFs could help differentiate patients with LV function recovery after TAVI from patients with persistent hemodynamic dysfunction.

## 1. Introduction

Aortic stenosis (AS) is the most common valvular disease needing intervention [1]. Transthoracic echocardiography is the cornerstone of the diagnosis and stratification of the disease [2], with multimodal and integrated evaluation being reserved for those cases not clearly defined with echo, in particular low-flow low-gradient AS [3,4,5].

Treatment of severe AS is currently indicated in symptomatic patients, in those who are asymptomatic with impairment of LV systolic function without other cause, and/or for patients with very severe disease [6]. Both surgical and percutaneous intervention are currently indicated for the treatment of the disease. Transcatheter aortic valve implantation (TAVI) has recently been established as the option of choice for the treatment of older patients or for those with high surgical risk [7,8,9,10,11], whereas aortic valve replacement (AVR) is preferred for younger patients due to concerns regarding the durability of prosthetic valves, albeit TAVI is proven to be non-inferior in these patients at shorter follow-up [12,13,14].

Interest in the analysis of intracardiac hemodynamic forces (HDFs) with echocardiography has grown in recent years, due to the potential to evaluate more accurately LV function and to indirectly estimate its deformation during the cardiac cycle [15]. HDFs are computed by evaluating tissue position, velocity and acceleration at the endocardial border by means of Navier-Stokes equation (Online Appendix A), values already used for the calculation of strain and strain rate. Alterations of HDF precede those of classical echocardiographic parameters; therefore HDF analysis holds the potential to identify pathological changes early and at a subclinical stage.

This work aims to assess HDF changes in patients undergoing TAVI for severe aortic stenosis to highlight possible adaptation of the LV to the new hemodynamic conditions after the intervention. Indeed, the acute reduction in afterload after the procedure has an effect on the contraction dynamic of the left ventricle and on the long-term positive remodeling.

## 2. Materials and Methods

Consecutive patients with severe aortic stenosis referred to our center for TAVI were prospectively enrolled between March 2021 and September 2022. All patients willing to sign the written informed consent form were included in the analysis. Patients with previous surgical valve replacement and/or repair were excluded.

Standard echocardiography examinations, according to current recommendations [16,17], were performed with EPIQc7 (Philips Healthcare) machine equipped with a X5-1 transducer, before and after TAVI. The post-procedure echocardiographic evaluation was performed as soon as possible after patient mobilization. The variability in this time is due to the fact that the echocardiographic evaluation was performed when the patients’ clinical conditions were stable (defined as the absence of active bleeding and unresolved vascular access complications, and euvolemic status). All the exams were performed by the same experienced cardiologists (authors AV, LZ and GA). The LV 2D global longitudinal strain (GLS) was quantified using new software Medis Suite Ultrasound (Medis Medical Imaging Systems, Leiden, The Netherlands). Using the same software, we were able to obtain HDFs through the knowledge of the LV geometry, endocardial velocities, obtained by speckle-tracking (ST), plus the area of the aortic and mitral orifices, carefully calculated by drawing the internal diameter of the valve’s annulus from the parasternal long axis-view [18]. Several HDF parameters were collected and classified in three groups as detailed in the three subsections below.

### 2.1. Amplitude Parameters

LV longitudinal force (LVLF) as the mean amplitude of the longitudinal force throughout the cardiac cycle; since it includes both positive and negative values, the amplitude was computed as the root mean square of all values.LV systolic longitudinal force (LVsysLF), calculated as for the LVLF above, but limited to the systolic phase only.LV impulse (LVim) as the mean longitudinal force during the systolic propulsive phase, when the force is positive (directed from the LV cavity toward the aorta); it is the area under the curve of the positive force profile during systole, normalized by the corresponding time interval [19].LV suction (LVs) as the mean longitudinal force during the period following propulsion while the force is negative, which is computed as the LVim but in the period comprising the end of the systole (when the force decelerates the exiting flow, with the aorta open and the mitral valve closed) and the initial part of the diastole (the effective suction when the mitral inflow accelerates, with the aorta closed and the mitral valve open).

### 2.2. Timing Parameters

Time from R-wave to positive peak of systolic LV longitudinal force, including the rates of force generation and force decay (RtoPeak).Duration of LV negative longitudinal force in the transition from systole to diastole.Time from the start of relaxation to positive peak of diastolic LV longitudinal force.

### 2.3. Orientation Parameters

Ratio between the transverse force and the longitudinal force (TF/TL).Dominant angle of the force vector, ranging from 90° (when the force is perfectly parallel to the base-apex axis) to 0°.

Reference values of normality were derived from those of healthy subjects reported in the literature [20]. A detailed description of the principles at the basis of echocardiographic evaluation of HDFs is provided in the Appendix A. Hemodynamic forces are evaluated from the results of speckle tracking and the uncertainty of the derived parameters is mainly imputable to that of speckle tracking. This uncertainty is mitigated when computing global parameters that combine tracking at all points in a single integral measure. Numerous studies have demonstrated that global strain parameters derived from speckle tracking present a level of uncertainty that is comparable to that of other clinical parameters. Similarly, the reproducibility of the hemodynamic force metrics, which is another global property derived from speckle tracking, was addressed in recent studies that reported reproducibility results analogous to those of global strain parameters [20,21,22,23].

In addition to routine echocardiographic examination, the following parameters were also collected: RV dimensions, fractional area change (FAC), RV free wall strain, RV endo strain-rate, LV inward displacement (% of LV walls displacement towards the LV center), LV endo strain-rate, LV stroke work (SW) with pressure-volume loop and hemodynamic work.

### 2.4. Statistical Analysis

Statistical Package for Social Sciences (SPSS Inc., Chicago, IL, USA) was used for statistical analysis. Continuous variables were reported as mean (±standard deviation). Categorical variables were reported as percentages. We tested for normality with Kolmogorov–Smirnov test with not significant results for all variables. Comparisons between pre- and post-TAVI groups were performed by means of a paired Student’s *t*-test for continuous variables.

Patients were then divided into subgroups, according to the presence of coronary artery disease (CAD), other cardiopathy (e.g., cardiac amyloidosis), paravalvular leak after TAVI, gender and subtype of aortic stenosis (NF-NG, LF-LG and paradoxical). Analysis of variance (ANOVA) and the Mann–Whitney test were used for multigroup comparison of continuous variables. The categorical variables were compared using the chi-squared test. A 95% confidence interval (CI) was adopted and a *p*-value < 0.05 was considered statistically significant.

## 3. Results

Twenty-five patients fulfilling the study inclusion criteria undergoing TAVI were prospectively enrolled at our center. Post-procedure echocardiographic evaluation was performed 2.4 (±1.06) days after the procedure.

The demographic, clinical and echocardiographic characteristics of the study population are reported in Table 1. Mean age was 83 ± 5 years and nearly two thirds of the patients were male (64%). High-gradient AS was the most common subtype of severe AS (84%), followed by low-flow low-gradient AS with reduced EF (12%); only one (4%) patient had low-flow low-gradient AS with preserved EF.

Other valvopathies were found in a significant portion of the population, with at least moderate mitral regurgitation in eight (32%) patients, at least moderate aortic regurgitation in nine (36%) patients and at least moderate tricuspid regurgitation in eight (32%) patients. Conduction disturbance was present in nine (36%) patients at baseline.

Table 2 summarizes the mean difference of the examined echocardiographic parameters before and after the procedure. The LVLF apex-base values were significantly higher after the procedure (mean difference 1.79%; 95% CI 1.07–2.5; *p*-value < 0.001) (Figure 1). Statistically significant variations of LVsysLF apex-base (mean difference 2.6%; 95% CI 1.57–3.7; *p*-value < 0.001) and LVIm (mean difference 2.9%; 95% CI 1.48–4.3; *p*-value < 0.001) were observed (Figure 2). In the post-TAVI group, the LVLF angle (mean difference 1.5°; 95% CI 0.07–2.9; *p*-value = 0.041) and LVim angle (mean difference 2.16°; 95% CI 0.76–3.56; *p*-value = 0.004) were significantly higher. An intergroup significant difference was also found with regards to flow rate and E/e’ avg. (*p*-value = 0.003 and *p*-value = 0.026, respectively).

After the procedure, a normalization of LVLF apex-base, LVsysLF apex-base and LVim apex-base values was observed in a greater percentage of patients, while not for the LV GLS and LVEF parameters (Table 3).

Finally, for the subgroups analysis, no statistically significant differences were found for the variables of interest.

## 4. Discussion

Our study has demonstrated that echocardiographic analysis of HDFs is feasible and can detect early adaptations to the new hemodynamic conditions after TAVI. In particular, the systolic components of hemodynamic forces (LVsysLV and LVim), as well as total hemodynamic force (LVLF), improved significantly after the procedure, highlighting how contractility recovery is related to a marked improvement in intraventricular fluid dynamics. This virtuous circle leads to an improved outward flow across the left ventricular outflow tract (LVOT), as demonstrated by significantly higher FR and lower LV filling pressure expressed by E/e’ average. Conversely, in our cohort, we observed only a marginal improvement in GLS while there was no substantial difference regarding LVEF. An increase in suction component of HDFs, albeit not significant, probably due to the presence of patients with paravalvular leak after procedure, was also observed. Not only amplitude parameters but also orientation parameters of HDFs were ameliorated after TAVI. After the procedure, the direction of LVLF and systolic components of HDFs tend to normalize as they approach 90°, with a marked reduction in transversal components (Figure 3).

TAVI has been established as the treatment of choice in patients with severe AS at high risk for surgery [6]. Furthermore, more recently the PARTNER 3 and Evolut Low Risk trials demonstrated that TAVI is non-inferior to AVR in low-risk patients at two-year follow-up [12,13].

Previous studies had already evaluated the effects of TAVI on the classical echocardiographic parameters of LV systolic function. It had been demonstrated that GLS improved after the procedure during the follow-up, and that this effect was maintained at one year [24,25]. However, little was known about the acute echocardiographic predictors of post-procedure recovery. In this optic, HDF evaluation holds the potential to highlight acute change and possibly predict improvement. Indeed, a quantitative analysis of the dynamic segmental contraction–relaxation sequence of the heart underscores the importance of mechanical synchrony and synergy in the hemodynamic performance of the normal and pathological LV. Blood flow analysis may provide insights into cardiac physiology, unachievable with conventional cardiovascular imaging [26]. It has already been demonstrated that deformation analysis with GLS and strain rate is superior for morphologic and functional characterization of LV function [27]. The analysis of intracardiac HDFs, corresponding to the global value of intraventricular pressure gradient (IVPG), which drive the blood flow in the heart, offers a rigorous method to explore IVPGs and blood flow within LV [28]. In our cohort, the absence in LVEF improvement confirms this finding, the early effects of the procedure did not induce LV remodeling and improvement in classical markers of systolic function. This represents a spatial–temporal course of the pressure gradients generated by the cyclical movement of the blood and tissue boundary, and it is therefore considered that the fluid dynamics correlate to deformation imaging. An innovative feature inherent in HDF analysis is the possibility of exploring the time course of the HDF curve during the heartbeat [15].

This is particularly interesting in our cohort, where, of the five patients who underwent a PM implantation after TAVI, three did not present any conduction disturbances at baseline ECG, but an alteration in the profile of the HDF curve was already present before the procedure. In particular, a double systolic peak, similar to that present in patients with mild desynchrony (Figure 4), suggests that, along with classical risk factors [29], HDF analysis may be useful to identify those patients at higher risk of PM implantation and therefore needing stricter ECG monitoring. The definite parameters associated precisely with specific cardiac disorders still need to be validated, but this approach clearly holds the potential for a deeper and more personalized analysis.

### Limitations

This study presents some limitations. This single-center, single-ethnicity study, with heterogeneous patient characteristics and a small size, does not allow the generalization of the results to the entire population. Moreover, the current approach should be considered an estimation of hemodynamic forces since the analysis is dependent on 2D image quality and frame rates. Finally, the lack of a clinical follow-up prevents the assessment of a possible association between early hemodynamic effects and long-term effects on LV function. Indeed, the majority of our patients presented with high-gradient normal-flow aortic stenosis, which is not usually associated with LVEF reduction. The results of our analysis may therefore be limited to this specific population of preserved LVEF; on the other hand, there is a strong rationale; GLS had already improved early after the procedure and suggest that changes in HDF could predict long-term remodeling.

The subgroup analyses are indeed limited by the small sample size; the subgroups were chosen because they all influence cardiac hemodynamics and the LV contraction sequence. Therefore, subgroup analyses were performed as hypothesis generating, in order to observe possible relationships needing further investigation, as in the case of abnormal HDF curve and subsequent PM implantation. However, aim of the study was to evaluate the feasibility of adopting HDFs in routine clinical practice and to provide new elements for the stratification of patients undergoing TAVI.

## 5. Conclusions

HDFs improve early after TAVI in patients with severe aortic stenosis, both in their amplitude and orientation parameters. Conversely, no significant differences were found in GLS and LVEF before and after the procedure.

The clinical role for HDFs in valvular disease is still under investigation; however our findings strongly suggest a possible role for their use in the early evaluation of patients undergoing percutaneous interventions and pave the way for a new area of research.

## Figures and Tables

**Figure 1 jcm-12-01218-f001:**
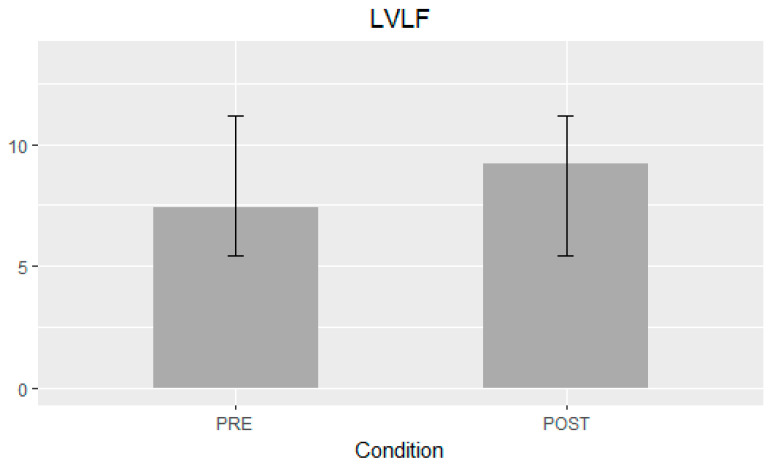
Left ventricular longitudinal force (LVLF) apex-base mean difference ± SD.

**Figure 2 jcm-12-01218-f002:**
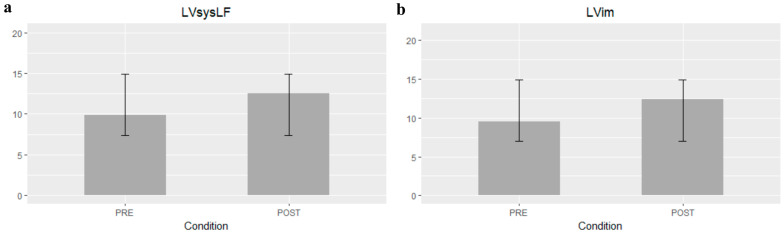
Left ventricular systolic longitudinal force (LVsysLF) apex-base (**a**) and Left ventricular impulse (LVim) apex-base mean differences ± SD (**b**).

**Figure 3 jcm-12-01218-f003:**
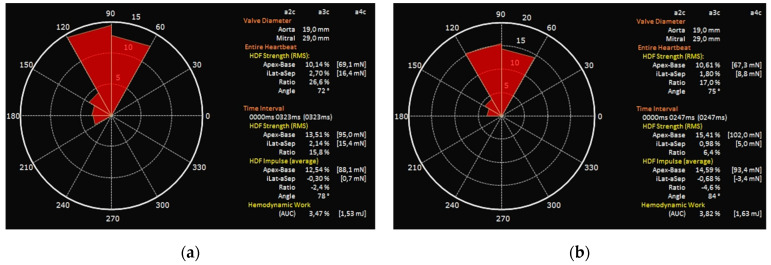
LVim intensity-weighted polar histogram before and after TAVI. Distribution and intensity of the left ventricular hemodynamic forces during systolic impulse are shown as red isosceles triangles within a polar histogram. (**a**) Patient before TAVI with resultant angle of 78° (**b**) Patient after TAVI with reduction in transverse components resulting in an angle of 84°.

**Figure 4 jcm-12-01218-f004:**
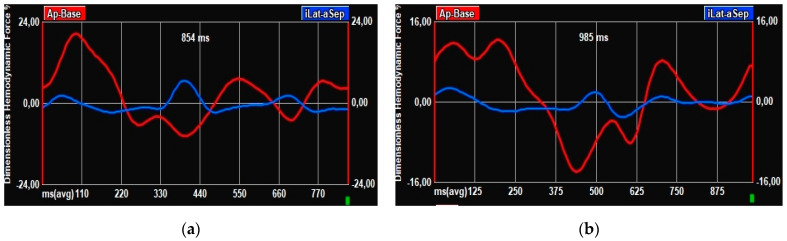
HDF curves. (**a**) HDF curve in patient without baseline conduction disturbances and without PM implantation after TAVI. (**b**) HDF curve in patient without conduction disturbances at baseline ECG but undergoing PM implantation post-TAVI.

**Table 1 jcm-12-01218-t001:** Demographic, clinical and echocardiographic characteristics.

Variable	Mean (±SD) or N (%)
Age—years	83 ± 5
Female sex—no. (%)	9 (36%)
Body surface area—m^2^	1.79 ± 0.22
CAD—no. (%)	9 (36%)
Other cardiopathy—no. (%)	5 (20%)
Conduction disturbances—no. (%)	9 (36%)
HG AS—no. (%)	21 (84%)
LF-LG AS with reduced EF—no. (%)	3 (12%)
LF-LG AS with preserved EF—no. (%)	1 (4%)
Moderate or severe MR—no. (%)	8 (32%)
Moderate or severe AR—no. (%)	9 (36%)
Moderate or severe TR—no. (%)	8 (32%)
Left ventricular ejection fraction (LVEF)—%	57 ± 8
LV global longitudinal strain (GLS)—%	−19 ± 4.5
Endo strain rate (SR) LV—no.	−0.87 ± 0.28
Stroke volume index (SVi)—mL/m^2^	43 ± 8
Flow rate (FR)—mL/s	234 ± 50
Hemodynamic work—(mJ)	0.89 ± 0.97
E/e’ average—no.	15 ± 5
Left ventricular longitudinal force (LVLF) apex-base—%	7.41 ± 2.65
Left ventricular systolic longitudinal force (LVsysLF) apex-base—%	9.85 ± 3.15
Left ventricular impulse (LVim) apex-base—%	9.52 ± 2.95
Left ventricular suction (LVs) apex-base—%	−4.92 ± 2.3

AR: Aortic Regurgitation; CAD: Coronary Artery Disease; EF: Ejection Fraction; HG AS: High-Gradient Aortic Stenosis; LF-LG AS: Low-flow Low-Gradient Aortic Stenosis; MR: Mitral Regurgitation; TR: Tricuspid Regurgitation.

**Table 2 jcm-12-01218-t002:** Paired *t*-test for continuous variables before and after TAVI.

Variable	Mean Δ	C.I. 95%		*p*-Value
		Lower Limit	Upper Limit	
LVLF apex-base (%)	1.80	1.07	2.52	<0.001
LVsysLF apex-base (%)	2.64	1.57	3.70	<0.001
LVim apex-base (%)	2.89	1.48	4.29	<0.001
LVLF angle (°)	1.52	0.07	2.97	0.04
LVim angle (°)	2.16	0.76	3.56	0.004
FR (ml/s)	31.40	12.19	50.61	<0.001
E/e’ avg.	−1.42	−2.64	−0.19	0.03
LVs apex-base (%)	−0.54	−1.31	0.22	0.16
LVs angle (°)	0.16	−3.31	3.63	0.93
SVi (mL/m^2^)	0.36	−3.43	4.16	0.84
LVEF (%)	1.44	−0.15	3.03	0.07

FR: Flow Rate; LVim: Left ventricular impulse apex-base; LVLF: Left ventricular longitudinal force apex-base; LVs: Left ventricular suction apex-base; LVsysLF: Left ventricular systolic longitudinal force apex-base; Svi: Stroke volume index.

**Table 3 jcm-12-01218-t003:** Percentage of patients with normal values of each parameter before and after procedure. LVLF, LVsysLF and LVim normal cut-offs derived from reference age-related values in patients with normal LV function [20]. LVEF considered normal if ≥52% for male and ≥54% for female sex. LV GLS considered normal if ≥−20%.

Variable	% of Patients within Normality Pre-TAVI	% of Patients within Normality Post-TAVI
LVLF apex-base (%)	4	56
LVsysLF apex-base (%)	4	52
LVim apex-base (%)	4	60
LVEF (%)	72	72
LV GLS (%)	56	60

LVEF: Left ventricular ejection fraction; LV GLS: Left ventricular Global Longitudinal Strain; LVim: Left ventricular impulse apex-base; LVLF: Left ventricular longitudinal force apex-base; LVsysLF: Left ventricular systolic longitudinal force apex-base.

## Data Availability

The study data will be made available upon request to the corresponding author.

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
