# Peer review of "Acute Modification of Hemodynamic Forces in Patients with Severe Aortic Stenosis after Transcatheter Aortic Valve Implantation"

_jcm, 2023, doi:10.3390/jcm12031218_

Round 1
Reviewer 1 Report
The authors presented a study on how hemodynamics forces evolve in patients before and after TAVI.
However, even tough the study tackles a relevant topic, there is a few points, which could be improved, that I would like to highlight.
Introduction
Line 52. The authors have added supplementary material to better explain how they compute hemodynamics forces. However, it would be help clarify the paper if they could add some of this description to the introduction. How do they derive them? Have they been validated for other pathologies?
Also, they authors could better explain why they want to characterize the hemodynamic forces. Do they want to use them to assess the success of the treatment? and if so, could they please details what metrics do physicians use to assess the effect of the TAVI, and consequently, how using hemodynamics forces could help?
Material and Methods
Line 109. The authors detailed that they tested the normality of the data distribution and that depending on the results, they present data as mean +/- SD or median interquartile range. Could the authors list the data distributions that were found to be not normally distributed? It looks like, that in the tables, they always present the data with their mean +/- SD (or they use the paired t-test which suggests all metrics are normally distributed).
Line 115. The authors have performed an analysis of different subgroups. However, their number of patients is not very large, and some groups have only a few patients. Could the authors explain why they perform all these subgroup analyses? are they all relevant? Maybe the authors should focus the most relevant group, for a better clarity of the manuscript. Especially as line 157, they conclude that their subgroup analysis didn’t show any differences.
Results and Discussion
Line 124. The time of the echocardiographic evaluation was 2.4 +/- 1.06. Could the authors justify the choice of this time post-surgery for the evaluation? Also, could the variability in this time post-surgery results in variability in the measured metrics (i.e. how fast are the changes post-surgery?)? (which goes with the limitations written line 224).
Figure 1 and Figure 2. Could the authors plot the results in the format mean +/I IC or STD?
Line 169. The authors state that the contractility recovery is related to a marked improvement in intraventricular fluid dynamics. However, they also state that they observe no substantial difference regarding LVEF. Could the authors explain why they obtained these somehow contradictory results?
Line 223. The hemodynamics forces are derived from US measurements, which tend to be noisy. Have the authors any estimation of the “uncertainty” of these metrics? This could help strengthening their results (if the uncertainty is much lower than the difference they observe).
Minor comments
Line 127. The sentence “Mean age was […]” is unclear, please rephrase it.
Author Response
Reviewer 1
The authors presented a study on how hemodynamics forces evolve in patients before and after TAVI. However, even tough the study tackles a relevant topic, there is a few points, which could be improved, that I would like to highlight.
We thank the Reviewer for her/his comments which allowed us to greatly improve our original work.
Introduction
Line 52. The authors have added supplementary material to better explain how they compute hemodynamics forces. However, it would be help clarify the paper if they could add some of this description to the introduction. How do they derive them? Have they been validated for other pathologies?
We thank the Reviewer for her/his comment, that allowed to improve the readability of the paper. A brief description of how HDF are derived has been added to the Introduction section.
Also, they authors could better explain why they want to characterize the hemodynamic forces. Do they want to use them to assess the success of the treatment? and if so, could they please details what metrics do physicians use to assess the effect of the TAVI, and consequently, how using hemodynamics forces could help?
We thank the Reviewer for her/his comment, which allowed to clarify an important aspect of our research. The characterization of hemodynamic forces after TAVI is not intended to assess the success of the treatment; in fact, other aspects are used to evaluate the acute procedural success (e.g. reduction in mean transvalvular pressure gradient and absence of prosthetic leak), whereas HDFs assessment holds the potential to identify early cardiac adaptations to the new hemodynamic condition and, possibly, predict positive long term remodeling. To better clarify this aspect a sentence has been added at the end of the Introduction.
Material and Methods
Line 109. The authors detailed that they tested the normality of the data distribution and that depending on the results, they present data as mean +/- SD or median interquartile range. Could the authors list the data distributions that were found to be not normally distributed? It looks like, that in the tables, they always present the data with their mean +/- SD (or they use the paired t-test which suggests all metrics are normally distributed).
We thank the Reviewer for this comment, and we apologize if this aspect was not clear in the original manuscript. We tested for normality with Kolmogorov–Smirnov test with not significant results for all variables. In order to avoid any possible misreading the statistical analysis section has been modified.
Line 115. The authors have performed an analysis of different subgroups. However, their number of patients is not very large, and some groups have only a few patients. Could the authors explain why they perform all these subgroup analyses? are they all relevant? Maybe the authors should focus the most relevant group, for a better clarity of the manuscript. Especially as line 157, they conclude that their subgroup analysis didn’t show any differences.
We thank the Reviewer for this relevant comment. We agree that small sample size is a limitation of our study as already reported in the Limitations section of the manuscript; however, these subgroups were chosen because they all influence cardiac hemodynamics and LV contraction sequence. Therefore, subgroup analyses were performed as hypothesis generating, in order to observe possible relationships needing further investigation, as the case of abnormal HDF curve and subsequent PM implantation (Figure 4). In order to deal with this important aspect a sentence has been added to the Limitations section.
Results and Discussion
Line 124. The time of the echocardiographic evaluation was 2.4 +/- 1.06. Could the authors justify the choice of this time post-surgery for the evaluation? Also, could the variability in this time post-surgery results in variability in the measured metrics (i.e. how fast are the changes post-surgery?)? (which goes with the limitations written line 224).
We thank the Reviewer for her/his comment which allowed us to clarify this important aspect of our research. The post-procedure echocardiographic evaluation was performed as soon as possible after patient mobilization. The variability in this time is due to the fact that the echocardiographic evaluation was performed when the patients’ clinical conditions were stable (defined as absence of active bleeding and unresolved vascular access complications, and euvolemic status). In order to clarify this aspect a sentence has been added to the Methods section.
Figure 1 and Figure 2. Could the authors plot the results in the format mean +/I IC or STD?
We thank the Reviewer for her/his comment which allowed us to improve the readability of our work. Figure 1 and Figure 2 have now been amended.
Line 169. The authors state that the contractility recovery is related to a marked improvement in intraventricular fluid dynamics. However, they also state that they observe no substantial difference regarding LVEF. Could the authors explain why they obtained these somehow contradictory results?
We thank the Reviewer for her/his comment. The absence of improvement in LVEF was, somehow, expected as the early effects of the procedure did not induce already LV remodeling and improvement in classical marker of systolic function. On the other end the acute effect on HDF were already visible, as these measures directly depends on the hemodynamic status. In order to explain this point, the Discussion and Limitations sections have been broadened.
Line 223. The hemodynamics forces are derived from US measurements, which tend to be noisy. Have the authors any estimation of the “uncertainty” of these metrics? This could help strengthening their results (if the uncertainty is much lower than the difference they observe).
We thank the Reviewer for her/his comment which allowed us to clarify an important aspect of our work. We agree that echocardiographic measurements present an intrinsic level of uncertainty due to variability in image acquisitions; a “uncertainty quantification” for analysis in the clinical setting is commonly performed by assessing reproducibility.
Hemodynamic forces are evaluated from the results of speckle tracking and the uncertainty of the derived parameters are mainly imputable to that of speckle tracking. This uncertainty is mitigated when computing global parameters that combine tracking at all points in a single integral measure. Numerous studies have demonstrated that global strain parameters derived from speckle tracking present a level of uncertainty that is comparable to that of other clinical parameters. Similarly, the reproducibility of the hemodynamic force metrics, that is another global property derived from speckle tracking, was addressed in recent studies that reported reproducibility results analogous to those of global strain parameters [20-23].
In this work, we have further minimized this effect since all the exams were performed by the same experienced Cardiologists (authors AV, LZ and GA). The Methods section has now been modified in order to clarify this aspect.
Minor comments
Line 127. The sentence “Mean age was […]” is unclear, please rephrase it.
We thank the Reviewer for her/his comment and apologize for the typo. The sentence has now been corrected.
Reviewer 2 Report
Dear authors,
I would like to congratulate you on a well-conducted study and a well-written paper. The biggest concern I have regarding the study is a very small sample size. Although the results are statistically significant, there is an obvious concern regarding the applicability of the findings in other cohorts, especially since the study group consisted mostly of patients with preserved LVEF (72% of the population, as stated in the Table 3). Not surprisingly, the parameter that was within the normal range did not improve, but the parameter that was initially impaired improved in this short post-TAVI observation. In my opinion, these findings should rather be discussed as early changes that can be observed in patients with preserved LVED rather than a competitive parameter in assessing early LV recovery after TAVI.
Author Response
Dear authors,
I would like to congratulate you on a well-conducted study and a well-written paper. The biggest concern I have regarding the study is a very small sample size. Although the results are statistically significant, there is an obvious concern regarding the applicability of the findings in other cohorts, especially since the study group consisted mostly of patients with preserved LVEF (72% of the population, as stated in the Table 3). Not surprisingly, the parameter that was within the normal range did not improve, but the parameter that was initially impaired improved in this short post-TAVI observation. In my opinion, these findings should rather be discussed as early changes that can be observed in patients with preserved LVED rather than a competitive parameter in assessing early LV recovery after TAVI.
We thank the Reviewer for her/his appreciation of our work and the comments. We agree that the fact LVEF did not improve was not surprising, as the early effects of the procedure did not induce already LV remodeling and most of the patients (84%) presented high gradient aortic stenosis, which is not usually associated with LVEF reduction. We completely agree with the fact that our findings are currently limited to patients with similar characteristics and further studies are warranted to broaden our observations; however, these results still strongly support a possible role of HDF evaluation early after TAVI in order to evaluate possible positive long-term remodeling. In order to clarify this aspect, a sentence has been added to the Discussion session and Limitations section has been broadened.